# Reducing Dog Relinquishment to Rescue Centres Due to Behaviour Problems: Identifying Cases to Target with an Advice Intervention at the Point of Relinquishment Request

**DOI:** 10.3390/ani11102766

**Published:** 2021-09-22

**Authors:** Natalie Powdrill-Wells, Sienna Taylor, Vicky Melfi

**Affiliations:** 1Wood Green the Animals Charity, Godmanchester PE29 2NH, UK; 2Department of Animal and Land Sciences, Hartpury University, Gloucester GL19 3BE, UK; Sienna.Taylor@hartpury.ac.uk (S.T.); Vicky.Melfi@hartpury.ac.uk (V.M.)

**Keywords:** behaviour, domestic dog, relinquishment, rescue centre

## Abstract

**Simple Summary:**

Annually, thousands of dogs are relinquished to rescue centres globally. Dog owners report that a leading cause for relinquishment are their dogs’ behavioural problems. Efforts are needed to reduce dog relinquishment, by enabling dog owners to feel comfortable and confident with having them in their home. Free behavioural advice was offered to 1131 dog owners at the time of them calling to relinquish their dog to animal welfare charity. Behavioural advice was accepted by 24.4% of the dog owners calling to relinquish their dogs. The advice was accepted almost six times more often by owners with dogs with general management behaviour problems, compared to owners who had problems with aggression between dogs in their home. Offering free behavioural advice reduced the number of dog relinquishments. Consequently, the stress associated with maintaining a dog in sheltered conditions and rehoming them, and the trauma for the owner was eliminated in a quarter of cases. Moving forward, additional strategies are needed to enable more owners to keep their dogs in their homes and reduce relinquishment.

**Abstract:**

Behaviour problems are a leading reason for dogs being relinquished to rescue centres across the world every year. The aim of this study was to investigate whether free behavioural advice would be accepted at the point of an owner requesting to relinquish their dog for behavioural reasons. The call records of 1131 relinquishment requests were reviewed and analysed to establish if the offer of free behaviour advice was accepted. The results showed that advice was accepted in 24.4% of relinquishment requests and behavioural problem was a significant predictor of whether advice was accepted (*p* < 0.001). The odds of advice being accepted were 5.755 times (95% CI: 2.835–11.681; *p* < 0.001) greater for a relinquishment request due to problems with general management behaviours compared to aggression between dogs in the home, representing 4.2% and 20.2% of overall relinquishment requests. These data suggest that owners are prepared to accept behaviour advice at the point of relinquishment request, so advice interventions could have potential to impact the levels of dog relinquishment to rescue centres. The impact of an intervention offering behaviour advice may be limited by overall levels of advice acceptance by owners and therefore complimentary proactive solutions to reduce behavioural relinquishments should also be considered.

## 1. Introduction

Every year thousands of dogs are relinquished by their owners to rescue centres or animal welfare organisations (from here on referred to as rescue centres) throughout the United Kingdom (UK). An estimated 129,473 dogs entered UK rescue centres in 2009 and 48% of surveyed rescue centres were running at full capacity for the year [1]. These figures are over 10 years old, so this should be considered, but to the authors knowledge no more recent estimations exist. Companion animal relinquishment is a worldwide problem, for example every year nearly 4 million dogs are estimated to enter rescue centres in the United States (US) [2], more than 100,000 dogs enter rescue centres in Spain [3] and over 200,000 admitted to rescue centres in Australia [4]. In 2009, 75% of dogs relinquished to rescue centres in the UK were matched with new owners and 10.4% were euthanised [5]. Aside from the risk of euthanasia, relinquishment to a kennel compared to the home environment exposes dogs to a number of stressors including social isolation [6,7], increased noise levels [8], spatial restriction [9,10] and loss of control and predictability [11], all of which may negatively impact welfare [12].

Problematic behaviours displayed by dogs are one of the most frequent reasons for relinquishment worldwide. Carter and Taylor [13] found that dogs in the Sunshine Coast region of Australia were primarily relinquished due to behavioural reasons (15%) and in a US based study Kwan and Bain [14] found behaviour problems played at least some part in 65% of owner’s decision to relinquish their dog. In a UK study by Diesel et al. [15], 34.2% of dogs were relinquished to rescue centres due to the owners’ perception that the dog had behavioural problems. The most common problem behaviours cited for relinquishment were forms of aggressive or destructive behaviour [15,16,17]. Behavioural problems accounted for 65.6% of the dogs euthanised in UK rehoming centres in 2009 [1]. Relinquishment to a rescue kennel environment can in itself cause or exacerbate behaviour problems due to the stressors the dog is exposed to [18,19]. Behavioural problems often make dogs more difficult to rehome [19] and are associated with an increased risk of dogs being unsuccessfully adopted. The main reason for dogs being returned to the Dogs Trust in 2005 was behaviour problems (58.6%) [20].

The key to avoiding the negative impact of relinquishment on dog welfare, could be to prevent the leading causes of relinquishment and support owners to keep their dogs in their home [14,21,22]. Interventions to reduce dog relinquishment may be economically beneficial, as the costs may be small compared to the costs of caring for the dogs [20] and could allow funds to be diverted to address other welfare issues [23]. The body of primary research focusing on interventions to reduce dog relinquishment is limited [2,24], despite 75% of research articles relating to dog relinquishment recommending the introduction of interventions [24]. When owners seek advice for behaviour concerns it can have a protective effect against relinquishment; however, the majority of relinquishing owners (69.2%) do not seek advice for their dog’s behaviour problems [15]. Studies have suggested this could be because pet owner knowledge of the services available to help them retain their pet may be limited [18,21]. Where owners had not sought advice for dogs showing aggression towards people, their odds of returning the dog to the rescue centre were 11.1 times higher compared to dogs without behaviour problems, whereas if advice was sought, the odds were reduced to 5.6 times [20].

Research has suggested that many relinquishing owners would like to keep their dog if there was a form of support that might help resolve their problem [18,25]. Dolan et al. [21] found that 88% of owners approaching a rescue centre in Los Angeles, California for dog relinquishment chose to pursue services that could help them keep their dog. However, another US study experienced low uptake on free training support when this was made available to owners [26]. Understanding the factors that predict whether behaviour support will be accepted may enable rescue centres to target their resources effectively, which is especially important when lack of finances are one of the most often cited concerns by rescue centre workers in the UK [27]. When contacting rescue centres to relinquish their dog, owners could be reasonably expected to state the reason why they need to give up their dog before providing any further information on the dog. Therefore, understanding which behavioural problems owners are more likely to accept assistance for could be beneficial. This study aims to investigate whether the behaviour issue driving the relinquishment request can predict if an owner will accept behaviour advice to support them in retaining their dog.

## 2. Materials and Methods

This study retrospectively analysed call data relating to behaviour-based dog relinquishment requests to Wood Green, The Animals Charity [28]. Wood Green is an animal welfare and rehoming organisation based in Cambridgeshire, England. Wood Green operate an appointment-based intake system, meaning that individuals who want to relinquish a pet must call the centre to discuss before potentially being given an appointment. Wood Green offers free behavioural support to every individual calling to relinquish their dog for behavioural reasons. The data set runs from January 2017–August 2019. Wood Green granted permission for access to the data required for the study. Ethical approval for the study was granted by Hartpury University ethics committee (reference ETHICS2019-56).

Records relating to 1698 calls were made available for analysis in this study. All call records were anonymised prior to data processing. Due to the way the organisation recorded their call data, the data set that was analysed only contained single entries for each dog-owner dyad requesting relinquishment. Information collected from relinquishment request calls included: dog sex, neuter status, age, and breed; acquisition source of the dog; whether the offer of free behaviour support was accepted; and the reason for relinquishment request (RRR). All data were converted into categorical variables for analysis. Dogs were recorded as male or female and neutered or entire. Dog age was divided into 5 categories: <0.5 years, 0.5 years–<2 years, 2 years –<7 years, 7 years–<12years and ≥12 years. The ‘breed’ variable categorised the dogs into the 7 breed groups recognised by the Kennel Club [29] gundog; hound; pastoral; terrier; toy; utility; and working, plus a category for crossbreed. An additional variable, classified dogs as crossbreed or pedigree (as recognised by the Kennel Club). Source of dog acquisition was recorded in the following categories: family/acquaintance, internet/social media, breeder, Wood Green, rescue/charity, private sale, other/unknown (unknown was defined as owners reported not to know where the dog was from) and own litter. Whether behaviour advice was accepted was recorded dichotomously as yes or no. The RRR were divided into 11 categories: aggression towards owner; aggression around children; aggression around people; aggression around dogs; aggression between dogs in the home; anxious or obsessive behaviour; excessive vocalisation or hyperactivity; general management behaviours; inappropriate toileting; problems mixing with other animals; and separation or destruction related behaviours. The ‘general management behaviour’ category was comprised of management challenges, including recall problems, jumping up and pulling on the lead. ‘Mixing with other animals’ refers to behaviour with all animals except dogs.

The data were checked and cleaned in Microsoft Excel to evaluate for missing measures and duplicate records and then analysed using the statistical package SPSS (Version 26, 2020). For the statistical analysis, call records with extensive missing data were excluded, leaving a final sample size of 1131 records. Descriptive statistics were produced in Microsoft Excel and used to report the sex, neuter status, age, breed, acquisition source and RRR for all dogs and then separately for cases where behaviour advice was accepted and refused. A chi-square test was used to establish any associations between categorical variables. Binary logistic regression was used to establish a univariable association between the behaviour problem reported and whether the offer of free behaviour advice would be accepted. A multivariate logistic regression was then performed to establish whether there were any dog characteristics which predicted whether a dog would be presented for relinquishment for the behavioural problems where advice acceptance was most and least likely. Due to the chi-square test showing significant associations between many of the categorical independent variables, only dog age and sex were taken forward to the final multivariate regression. For the multivariate logistic regressions, age was reduced to 4 categories, with groups 7–<11 years and >12 years combined to ensure group sizes were statistically robust. The significance level for statistical analysis was set at *p* < 0.05.

## 3. Results

### 3.1. Demographic Data

Of the relinquishment request calls analysed as part of the study, 62.5% of the dogs were male (see Table 1) and 59% of dogs were neutered. When considered by sex, 53.5% of females and 62.2% of males were neutered. The most common age category within the sample was dogs from 2 years–<7 years (48.5%). Crossbreeds accounted for 52.7% of the sample and the most common Kennel Club breed group was terrier 16.4%, which compromised 34.8% of pedigree dogs in the study.

The most common reported sources of acquisition were family/acquaintance (22.6%), internet/social media (22.4%) and breeder (21.0%); however, when Wood Green and rescue/charity are combined to give an overall total of dogs originally from a rescue this becomes the most common source (282/1131, 24.9%).

### 3.2. Acceptance of Advice

The offer of free behavioural advice at the point of relinquishment request was accepted in 276/1131 cases (24.4%). Advice was accepted for (see Table 2): 26.7% of male dogs and 20.5% of female dogs; 27.7% of neutered dogs and 19.6% of entire dogs; 33% of dogs under 6 months old; 26.2% of crossbreeds and for 22.4% of pedigree dogs. The Kennel Club breed group where behaviour advice was accepted most was pastoral (29.6%) and least commonly was toy (14.6%). In terms of acquisition source, advice was accepted for 40.3% of dogs originally rehomed from Wood Green and 32.6% of dogs from other rescue centres/charities. The most common acquisition source: family/acquaintance saw advice accepted for 38/256 (14.8%) cases.

### 3.3. Behavioural Reasons

Aggressive behaviour between dogs in the home’ (20.2%) was the most common behavioural problem cited as the RRR (see Table 3). ‘Aggression around children’ also accounted for a large proportion (19.3%) of requests. The least common behavioural problems cited as RRR were inappropriate toileting (3.0%), anxious or obsessive behaviour (3.1%) and mixing with other animals (3.9%). When combined, aggressive behaviour towards people or other dogs accounted for 68.3% (773/1131) of RRR.

Behaviour advice was accepted most in cases of problems with ‘general management behaviours’ (42.6%), ‘anxious or obsessive behaviours’ (34.3%) and ‘aggression around dogs outside of the home’ (32.9%) (see Table 3). The univariate logistic regression model found RRR to be a significant predictor of whether behaviour advice would be accepted (*p* < 0.001). Dogs presented for relinquishment due to ‘general management behaviours’ had 5.755 times the odds (95% CI: 2.835–11.681; *p* < 0.001) of advice being accepted compared to dogs presented due to ‘aggression between dogs in the home’. All other RRR, except for ‘inappropriate toileting’ and ‘mixing with other animals’ had significantly higher odds of behaviour advice acceptance than ‘aggression between dogs in the home’. Dogs under the age of 6 months were not presented at all for ‘anxious or obsessive behaviours’ and dogs over 12 years were not presented for ‘anxious or obsessive behaviours’, ‘excessive vocalisation or hyperactivity’, ‘general management behaviours’ or ‘mixing with other animals’ (see Table 4).

Multivariate logistic regression modelling of the dog characteristic factors (see Table 5) which predict which dogs will be presented for ‘aggression between dogs in the home’ compared to other RRR, showed that both age and sex of dog were significant predictors. Female dogs had 1.739 times the odds (95% CI: 1.294–2.337; *p* < 0.001) of being presented for relinquishment due to aggression between dogs in the home than males. Dogs in age categories ‘0.5–<2 years’ (EXP(B): 0.649; 95% CI: 0.463–0.910; *p* = 0.012) and ‘>7 years’ (EXP(B): 0.583; 95% CI: 0.369–0.920; *p* = 0.021) showed lower odds of being presented for relinquishment due to ‘aggression between dogs in the home’ than dogs in age category ‘2–<7 years’.

For RRR ‘general management behaviours’, age contributed significantly to the multivariate logistic regression model, but sex did not (see Table 6). Compared with dogs aged ‘2–<7 years’, dogs under 6 months showed 9.904 times the odds (95% CI: 3.850–25.476; *p* < 0.001) and dogs aged ‘0.5–<2 years’ 2.659 times the odds (95% CI: 1.356–5.212; *p* = 0.004) of being presented for relinquishment due to ‘general management behaviours’ instead of alternative RRR.

## 4. Discussion

Dog behavioural problems can have far-reaching implications for the human–animal bond, interfering with everyday activities and creating frustration for owners [19]. Where owners are not satisfied, breakdown of the dog–owner bond can occur, leading to relinquishment of the dog [30].

### 4.1. Dog Demographics

In the current study, 62.5% of behaviour-based relinquishment requests were male dogs. Other studies have suggested that males make up a larger proportion of the rescue dog population than females [15,31]. Male dogs are also considered to be at greater risk than females for developing a range of behaviour problems [32]. Maybe these reasons are precursors to the fact that male dogs in the UK were found to have 1.4 times the odds of death due to an undesirable behaviour compared to females [33].

Neutered dogs accounted for 59% of relinquishment requests and in the general UK dog population neutered individuals represent between 57% [34] so this appears representative of the general population. Neutered dogs have been found to be more frequent in behavioural relinquishment categories [16], though the study was unclear how this related to the general dog population. Neutered dogs are considered at higher risk of developing many behavioural issues [32] and are relinquished more often for behaviour problems excluding aggression than entire dogs [31]. A UK study found that over half of relinquished dogs had been neutered but this figure may have been affected by the neutering policy of rescue centres as a percentage of the sample were returns [15], this is also the case with the current study.

Most of the dogs within this sample were over 2 years old but less than 7 years (48.5%). This differs from a UK study by Diesel et al. [15] where a relatively high proportion of relinquished dogs were under 6 months and a US study where significantly more relinquished dogs were younger than 2 years old [17]. Both studies looked at the spectrum of relinquishment reasons, rather than just behavioural relinquishment so this may partially explain the inconsistency. Dogs over 12 years, accounted for only 1.2% of this sample, this could be reflective of the suggestion that risk of relinquishment decreases with increasing length of ownership [17] or that older dogs are more likely to be euthanised instead of relinquished [35].

Crossbreeds appear to be overrepresented in this sample of behaviour-based relinquishment requests (52%) compared with the overall dog population in the UK (21.1%) [36]. Consideration should be given to the growing trend for designer crossbreeds over the past few years due to the source being several years old. Boyd et al. [33] found that crossbreeds were 1.39 times more likely to die before the age of 3 years due to undesirable behaviour compared to pedigrees. Conversely, Col et al. [32] found 58% of dogs presented to a behaviour clinic were pedigree and being pedigree or not was found not to be a significant factor for predicting most behaviour problems. When just pedigree dogs were considered, the terrier group was the most numerous (34.8%) in this study and terriers were overrepresented in comparison to their contribution to the overall UK pedigree dog population (20.4%) [36]. This supports the findings of Boyd et al. [33] who found terriers to be the most common breed group amongst dogs under 3 years whose death was attributed to undesirable behaviours. It is important to note that breed descriptions within this study were owner reported and not able to be confirmed by the researcher so may be inconsistent.

New et al. [17] found that dogs originally acquired from a rescue, pet shop, friend or as a stray were at an increased risk of relinquishment compared to those that the owner received as a gift. Other studies found that relinquished dogs were most commonly born with the owner [11] or acquired for free from friends or family [25]. Within this sample, the top three sources of acquisition were rescue (Wood Green and rescue/charity combined), family/acquaintance and internet/social media.

### 4.2. Advice Acceptance

The offer of free behavioural advice was accepted in just less than a quarter (24.4%) of cases in this study. Given that behavioural reasons account for a high percentage of dog relinquishment [15], this type of intervention represents an opportunity for a substantial reduction of behaviour-based dog relinquishment and could have a positive impact on the welfare of thousands of dogs. It is important to note that 567 call records were excluded from data analysis due to missing data which may have influenced the results. For these records, it is not possible to know why they were incomplete but if owner time constraints were a barrier, acceptance of advice to prevent relinquishment may have been unlikely. When compared to the uptake of other interventions designed to reduce pet relinquishment, the uptake rate in the current study sits in the middle of the range. Gunter et al. [26] found that 12.6% of owners attended group dog walks offering free access to a dog trainer whereas Dolan et al. [21] found that 88% of potential relinquishing owners chose to pursue support services over relinquishment. When vouchers offering financial assistance towards cat neutering were provided in the UK, 64.6% of owners redeemed their voucher [37]. The wider spectrum of relinquishment reasons were considered in these studies, meaning financial and practical support options were included, which may be perceived by owners to be an easier fix than following a treatment plan to work through a behaviour problem [33,38]. Diesel et al. [15] found that 30.8% of owners had sought advice for their dog’s behaviour problem before relinquishing to a rescue centre and Boyd et al. [33] found that where dogs had their death attributed to undesirable behaviour, only 12.9% of owners had sought advice to resolve the behaviour.

The current study has not investigated the effectiveness of the behaviour advice provided at preventing the relinquishment, but even if in all cases where advice was accepted, relinquishment was successfully prevented, 76.6% of cases could still result in relinquishment. This suggests that offering advice services at point of relinquishment request could make an important contribution to reducing relinquishment due to behavioural problems but will not suit all owner–dog dyads. A suite of services including free or low-cost dog training classes [39,40] and pro-active behaviour advice at veterinary appointments [41] may need to be considered to truly be effective at reducing behaviour-based relinquishment.

Dog owners are more likely to use sources of free support such as the internet than those with a cost [42]; however, even an intervention where free access to a trainer was provided had a low turnout [26]. The intervention involved group dog walks so those experiencing serious behaviour problems such as aggression may not have felt it was an appropriate setting to attend. Dog owners rated themselves more likely to look for free advice online or to call their vet than calling an animal rescue for free advice if they were experiencing problems with their dog [42]. Lack of awareness of services such as the one in this study, may contribute to the limited acceptance of advice. Weiss et al. [18] found more owners tried a helpline before relinquishment in an area where the helpline number was prominently displayed on the organisation’s website. Owners may have sought advice on the behaviour problem from an external source before approaching Wood Green and hence refused the offer feeling that it would be of no further help, unfortunately this data was not recorded in this study but would be interesting to consider in future research.

### 4.3. Behaviour Problems

Advice was most likely to be accepted for RRR ‘general management behaviours’ which included behaviours such as jumping up, pulling on a lead or recall issues. This category of behaviour problem was displayed by 4.2% of the dogs in this sample, so the overall impact on numbers of dogs relinquished would be limited. However, limited, reductions in relinquished dogs’ free up space and resources to other activities of the rescue centre.

Dog sex was not found to be a significant predictor of whether RRR would be ‘general management behaviours’ instead of other RRR. Dogs under the age of 2 years were significantly more likely to have this RRR than those aged from 2 years up to 7 years. Older dogs are considered to be calmer than younger dogs [43] which may explain the differences between the age groups. Younger dogs are also considered to be more trainable than older dogs [43] and therefore owners of younger dogs may be more willing to attempt to work through the problems they are experiencing.

Providing free or low-cost training classes may provide a more resource efficient way to reduce relinquishment due to perceived problems with a dog’s general management behaviours. Group sessions may be attractive to rescue centres who are often concerned about meeting the demand with limited resources [27]. A training class environment could create an opportunity for owners to get advice on behaviour problems whilst also providing opportunities for socialisation which in turn may reduce other behaviour problems. Studies have demonstrated that attending puppy training classes has the potential to reduce future levels of behaviour problems [40,44] and risk of future relinquishment [39].

Aggressive behaviours towards people or dogs contributed to over two thirds (68.3%) of the RRR in this study. Aggressive dog behaviour significantly decreases the chance of owners being very satisfied with their dog [30]. The single most common RRR within this study and the one which advice was least likely to be accepted for was ‘aggression between dogs in the home’. Aggression between pets was the second most common behaviour problem cited as the reason for death in dogs under three years in the UK [33] and aggression towards a familiar dog accounted for 65% of veterinary behaviour clinic cases in the US [45].

In the current study, female dogs had 1.739 times the odds of being presented for aggression towards dogs in the home over other RRR compared to males. Wrubel et al. [46] found that 68% of the cases of aggression between dogs in the household involved at least one female or a female–female pair. Similarly, 70% of dog pairs referred to a veterinary behavioural clinic for aggression between dogs in the home involved at least one female [47]. Dogs aged over 2 years, but less than 7 years had a higher likelihood of RRR ‘aggression towards dogs in the home’ instead of other RRR compared to dogs from 6 months up to 2 years and those over 7 years. The instigator of the aggression is often the younger individual [46] but the current study does not include data relating to this. Martinez et al. [48] suggest that the risk of inter-dog aggression increases with age and Casey et al. [49] supports this with specific reference to inter-dog aggression in the home. Whilst this partially supports the finding of the current study, it does not fit with the risk decreasing for those over 7 years, it may be that dogs of this age have already been relinquished at a younger age if showing this behaviour.

Research suggests that the attitude of the owner regarding attempts to resolve behaviour problems depends on how they perceive the problem [50]. Shore et al. [42] found that participants in their study were most likely to ask for behaviour advice if their dog displayed aggression towards people (7.26/10); this was the problem which participants also rated as the most serious. In the case of dogs not getting along with the owner’s other pets, the likelihood of asking for advice for the issue was rated as 6.42 out of 10, with 1 being ‘not at all likely’ and 10 being ‘extremely likely’. It is important to consider that in this study, all RRR are based on the owner’s perception of the behaviour and each has its own spectrum of seriousness which can present a challenge to clear comparisons. It is also possible that owners may be misrepresenting the behaviour to protect the dog [51] or providing a behavioural RRR as a socially acceptable reason [15] which may influence their likelihood to accept advice.

Owners may be less likely to choose to follow behaviour advice if they perceive that the training plan is too complex [38]. In the case of ‘aggression between dogs in the home’, it could be expected that training may require significant time investment, substantial financial cost and ongoing situational management from the owner which could decrease their likelihood of accepting advice. Advice was 3.814 times more likely to be accepted for aggression towards dogs outside of the home than aggression between dogs in the home, which suggests that owners may perceive barriers to tackling this behaviour within the home environment. Barriers to following behaviour advice relating to inter-dog aggression in the home could include the presence of children, inappropriate home set up or lack of time or finances. Data relating to the impact of these factors on advice acceptance were not recorded in the current study, but this warrants further investigation. Fifty-five percent of dogs presented for veterinary behaviour advice for aggression between dogs in the home had a poor outcome, of which over half had to be kept permanently separated [47], something which would require significant owner effort and changes to lifestyle. Other RRR within the current study which may significantly impact an owner’s home life such as inappropriate toileting and mixing with other animals were also amongst the RRR where owners were least likely to accept advice. Given the likelihood of a poor outcome following behaviour advice for a problem such as aggression between dogs in the home, consideration should be given to whether encouraging owners to attempt to resolve the problem could inadvertently negatively impact the welfare of the dogs involved and relinquishment may be the more appropriate option.

Based on owner’s suggested reluctance to accept advice for a serious but common problem such as aggression between dogs in the home, the key to reducing relinquishment for reasons such as this may be a proactive approach. Preventing behaviour problems from developing may also have additional benefits for dog and owner welfare as a situation with aggression between dogs in the home is likely to be extremely stressful. Some owners acquire a dog on an impulse [52] so providing advice to help well thought out decisions to be made about acquiring a dog may help to reduce some causes of relinquishment [53]. Rohlf et al. [54] found that 52.4% of highly committed dog owners reported giving the decision to get a dog a great deal of thought. Weiss et al. [55] reported that 27.3% of participants stated appearance was the single most important reason for choosing a dog, however after a few months of ownership the value of appearance dramatically reduces and behaviour and temperament are seen as much more important [18]. New owners often have to adjust their perception of dog ownership within the first few months of ownership [56] and a major theme amongst relinquishing dog owners is that they would devote more time, thought and planning to considering getting another dog in the future [57]. Care should be taken on providing advice for previous or current dog owners who consistently have greater odds of expecting benefits and reduced odds of expecting challenges with a new dog [58] and who may consider themselves experienced enough not to need to research further [52]. Offering temporary adoption programmes [59], which allow owners time to assess the relationship between new and existing dogs before officially adopting, may be beneficial in reducing relinquishment.

## 5. Conclusions

This study has shown that relinquishing dog owners are open to the offer of free behaviour advice at the point of relinquishment request, which may provide an important opportunity to potentially reduce behaviour-based dog relinquishment to rescue centres. The likelihood that free behaviour advice will be accepted at the point of relinquishment request can be predicted by the RRR. The effectiveness of this advice offering could be limited due to low uptake from owners for common but more complex issues. Proactive solutions encouraging attendance of training classes and promoting thorough research prior to choosing a new dog should be further explored as they may offer a more effective approach to reducing behaviour-based dog relinquishment in a greater number of cases. This study has also highlighted the need for future research to investigate the owner characteristics which influence the likelihood of behaviour advice being accepted and how effective behaviour advice is at preventing relinquishment when it is accepted.

## Figures and Tables

**Table 1 animals-11-02766-t001:** Descriptive statistics of calls received by Wood Green relating to behaviour-based dog relinquishment requests between January 2017 and August 2019.

Variable	Category	*N*	%
Dog sex	Male	707	62.5
	Female	424	37.5
Neuter status	Neutered	667	59.0
	Entire	464	41.0
Dog age	<0.5 years	39	3.4
	0.5–<2 years	368	32.5
	2–<7 years	548	48.5
	7–<12 years	162	14.3
	≥12 years	14	1.2
Breed group	Crossbreed	596	52.7
	Gundog	82	7.3
	Hound	42	3.7
	Pastoral	71	6.3
	Terrier	186	16.4
	Toy	48	4.2
	Utility	55	4.9
	Working	51	4.5
Crossbreed	Yes	596	52.7
	No	535	47.3
Dog acquired from	Family/Acquaintance	256	22.6
	Internet/Social media	253	22.4
	Breeder	238	21.0
	Wood Green	196	17.3
	Rescue/Charity	86	7.6
	Private sale	49	4.3
	Other/Unknown	28	2.5
	Own litter	25	2.2
Advice accepted	Yes	276	24.4
	No	855	75.6

**Table 2 animals-11-02766-t002:** Descriptive statistics for dog characteristics and whether advice was accepted or refused at the point of dog relinquishment request to Wood Green between January 2017 and August 2019.

		Advice Accepted			
		Yes		No	
Variable	Category	*N*	%	*N*	%
Dog sex	Male	189	26.7	518	73.3
	Female	87	20.5	337	79.5
Neuter status	Neutered	185	27.7	482	72.3
	Entire	91	19.6	373	80.4
Dog age	<0.5 years	13	33.3	26	66.7
	0.5–<2 years	99	26.9	269	73.1
	2–<7 years	126	23.0	422	77.0
	7–<12 years	34	21.0	128	79.0
	≥12 years	4	28.6	10	71.4
Breed group	Crossbreed	156	26.2	440	73.8
	Gundog	18	22.0	64	78.0
	Hound	12	28.6	30	71.4
	Pastoral	21	29.6	50	70.4
	Terrier	42	22.6	144	77.4
	Toy	7	14.6	41	85.4
	Utility	11	20.0	44	80.0
	Working	9	17.6	42	82.4
Crossbreed	Yes	156	26.2	440	73.8
	No	120	22.4	415	77.6
Dog acquired from	Family/Acquaintance	38	14.8	218	85.2
	Internet/FB	56	22.1	197	77.9
	Breeder	57	23.9	181	76.1
	Wood Green	79	40.3	117	59.7
	Rescue/Charity	28	32.6	58	67.4
	Private sale	9	18.4	40	81.6
	Other/Unknown	4	14.3	24	85.7
	Own litter	5	20.0	20	80.0

**Table 3 animals-11-02766-t003:** Descriptive and univariable logistic regression statistics for association between behavioural problem and acceptance of behavioural advice among dog behaviour-based relinquishment request calls to Wood Green, The Animals Charity between January 2017 and August 2019.

		All Dogs	Advice Yes	Advice No			
Variable	Category	*N (%)*	*N (%)*	*N (%)*	Exp(B)	95% CI	*p*-Value
Behaviour problem	Aggression between dogs in home	228 (20.2)	26 (11.4)	202 (88.6)	BASE		
	General management behaviours	47 (4.2)	20 (42.6)	27 (57.4)	5.755	2.835–11.681	<0.001
	Anxious or obsessive behaviour	35 (3.1)	12 (34.3)	23 (65.7)	4.054	1.806–9.100	0.001
	Aggression around dogs	82 (7.3)	27 (32.9)	55 (67.1)	3.814	2.061–7.058	<0.001
	Aggression towards owner	117 (10.3)	33 (28.2)	84 (71.8)	3.052	1.720–5.416	<0.001
	Aggression around people	128 (11.3)	36 (28.1)	92 (71.9)	3.040	1.734–5.330	<0.001
	Aggression around children	218 (19.3)	60 (27.5)	158 (72.5)	2.950	1.780–4.889	<0.001
	Separation or destruction related behaviours	115 (10.2)	30 (26.1)	85 (73.9)	2.742	1.531–4.913	0.001
	Excessive vocalisation or hyperactivity	83 (7.3)	20 (24.1)	63 (75.9)	2.466	1.290–4.715	0.006
	Mixing with other animals	44 (3.9)	7 (15.9)	37 (84.1)	1.470	0.595–3.634	0.404
	Inappropriate toileting	34 (3.0)	5 (14.7)	29 (85.3)	1.340	0.477–3.764	0.579

Note: There were statistically significant (<0.05) associations between all dog behaviour problems and whether behavioural advice was accepted, except those presented in Italic.

**Table 4 animals-11-02766-t004:** Descriptive statistics showing the count and percentage of each sex and age group category presenting for each behavioural problem amongst dogs presented for relinquishment due to behavioural reasons.

Variable	Category	Aggression between Dogs in the Home	Aggression around Children	Aggression around People	Aggression towards Owner	Aggression around Dogs	Anxious or Obsessive Behaviour	Separation Issues or Destruction Relatedissues	Excessive Vocalisation or Hyperactivity	General Management Behaviour	Mixing with Other Animals	Inappropriate Toileting
		*N* (%)	*N* (%)	*N* (%)	*N* (%)	*N* (%)	*N* (%)	*N* (%)	*N* (%)	*N* (%)	*N* (%)	*N* (%)
Sex	Male	118 (16.7)	139 (19.7)	101 (14.3)	84 (11.9)	52 (7.4)	22 (3.1)	73 (10.3)	52 (7.4)	29 (4.1)	16 (2.3)	21 (3.0)
	Female	110 (25.9)	79 (18.6)	27 (6.4)	33 (7.8)	30 (7.1)	13 (3.1)	42 (9.9)	31 (7.3)	18 (4.2)	28 (6.6)	13 (3,1)
Age	<0.5 years	8 (20.5)	3 (7.7)	1 (2.6)	6 (15.4)	1 (2.6)	-	1 (2.6)	5 (12.8)	8 (20.5)	2 (5.1)	4 (10.3)
	0.5–<2 years	62 (16.8)	53 (14.4)	43 (11.7)	41 (11.1)	14 (3.8)	11 (3.0)	43 (11.7)	48 (13.0)	24 (6.5)	24 (6.5)	5 (1.4)
	2–<7 years	131 (23.9)	113 (20.6)	63 (11.5	57 (10.4)	46 (8.4)	19 (3.5)	45 (8.2)	23 (4.2)	14 (2.6)	17 (3.1)	20 (3.6)
	7–<12 years	26 (16.0)	46 (28.4)	19 (11.7)	12 (7.4)	20 (12.3)	5 (3.1)	21 (13.0)	7 (4.3)	1 (0.6)	1 (0.6)	4 (2.5)
	≥12 years	1 (7.1)	3 (21.4)	2 (14.3)	1 (7.1)	1 (7.1)	-	5 (35.7)	-	-	-	1 (7.1)

**Table 5 animals-11-02766-t005:** Descriptive statistics and multivariate logistic model for dog characteristic variables associated with request for relinquishment due to aggression between dogs in the home among dogs presented for relinquishment due to behavioural problems.

		Aggression between Dogs in Home		Other RRR				
Variable	Category	*N*	%	*N*	%	Exp(B)	95% CI	*p*-Value
Sex	Male	118	51.8	589	65.2		BASE	
	Female *	110	48.2	314	34.8	1.739	1.294–2.337	<0.001
Age	<0.5 years	8	3.5	31	3.4	0.761	0.339–1.707	0.507
	0.5–<2 years *	62	27.2	306	33.9	0.649	0.463–0.910	0.012
	2–<7 years	131	57.5	417	46.2		BASE	
	>7years *	27	11.8	149	16.5	0.583	0.369–0.920	0.021

Note: statistically significant (*p* < 0.05) predictors denoted with *.

**Table 6 animals-11-02766-t006:** Descriptive statistics and multivariate logistic model for dog characteristic variables associated with request for relinquishment due to problems with general management behaviour among dogs presented for relinquishment due to behavioural problems.

		General Management Behaviours		Other RRR				
Variable	Category	*N*	%	*N*	%	Exp(B)	95% CI	*p*-Value
Sex	Male	29	61.7	678	63.1	BASE		
	Female	18	38.3	406	37.8	0.955	0.516–1.770	0.884
Age	<0.5 years *	8	17.0	31	2.9	9.904	3.850–25.476	<0.001
	0.5–<2 years *	24	51.1	344	31.7	2.659	1.356–5.212	0.004
	2–<7 years	14	29.8	534	49.3	BASE		
	7–<12 years	1	2.1	175	16.1	0.218	0.028–1.667	0.142

Note: statistically significant (*p* < 0.05) predictors denoted with *.

## Data Availability

Third Party Data. Restrictions apply to the availability of these data. Data were obtained from Wood Green, The Animals Charity and are available from the corresponding author with the permission of Wood Green, The Animals Charity.

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
