# Peer review of "Reducing Dog Relinquishment to Rescue Centres Due to Behaviour Problems: Identifying Cases to Target with an Advice Intervention at the Point of Relinquishment Request"

_animals, 2021, doi:10.3390/ani11102766_

Round 1
Reviewer 1 Report
This paper addresses an extremely important topic – that of whether providing free behavioral advice will mitigate surrender of pet dogs. It appears to be well done and the authors acknowledge limitations.
I am surprised that health issues (either the dog or the owner’s) were not provided as possible reasons for surrender.
Do not refer to dog gender (e.g., line 136, 144). Please find and replace with sex throughout. Gender is not appropriate for describing an animal’s sex as it is a psychological construct.
It would be important to know if the owners had already sought advice before calling to relinquish their dogs. It is possible that many more already tried obtaining advice that was unsuccessful. This is noted by the authors already in the discussion.
Although also impossible to determine from this existing data set, but the authors do not know how many calls were repeat calls rather than new instances so this should be noted.
The discussion is rather long for the scope of the data and could be streamlined a bit. However, I do appreciate their thorough consideration of caveats and possible explanations.
The paper is generally well written but there are some minor grammatical issues like the need for possessive apostrophes (e.g., line 293, 367, 396) and ; before “however” (e.g., line 70), and commas after clauses (e.g., line 74).
Lines 78, 80 “which” should be “that.”
Line 379, fix “of accept advice.”
Reviewer 2 Report
This is a good article and well worth publishing – it also has the potential to lead to further research in this field.
- The simple summary says “Annually, thousands of dogs are relinquished to rescue centres globally” – however the article itself only refers to UK and USA. Lines 43 and 44 say “Companion animal relinquishment is a worldwide problem, for example in the United States (US)”… (there is more to the world than UK and USA). Perhaps the researchers can find another reference for Asia, South America etc, or modify the wording. Ditto lines 51-52 that are footnoted to articles for studies in the US and Australia. If the articles themselves make this claim, maybe quote them – otherwise the reader may be left wondering.
- Parag lines 62-74 – do we know why owners do not seek advice? Is it because there is none readily available, or the owners do not know where to receive this advice. I make this comment because hubby and I have always had rescue dogs and if we had had behavioural problems we would not have known where to obtain advice (Australia), apart maybe from private dog trainers.
- Lines 87-89 “This study aims to investigate whether the presenting behavioural relinquishment reason can help to predict if an owner will accept behaviour advice to support them in retaining their dog at the point of relinquishment request.” Perhaps this can be reworded to be clearer. Also if you are saying that the reason for relinquishing a dog and its links to behavioural problems, may be an indicator whether the owners will accept advice, then it may be worthwhile to consider why owners do not seek advice – is it merely the dog’s behaviour or other reasons.
- Lines 87-99 – Great! Ethics approval has been given.
- I like the design of the project – very thorough.
- Lines 332-339 make a very good point.
- Lines 401-403 “Some owners acquire a dog on an impulse [50] so providing advice to help well thought out decisions to be made about acquiring a dog may help to reduce some causes of relinquishment” – yes! As I was reading this article, I wondered about the human element and its contribution to the issues raised in this article. And totally agree with the conclusion around lines 428-430 about future research.
Author Response
Response to Reviewer 2 Comments
Thank you for taking the time to review our article and providing positive comments and recommendations.
This is a good article and well worth publishing – it also has the potential to lead to further research in this field.
Point 1: The simple summary says “Annually, thousands of dogs are relinquished to rescue centres globally” – however the article itself only refers to UK and USA. Lines 43 and 44 say “Companion animal relinquishment is a worldwide problem, for example in the United States (US)”… (there is more to the world than UK and USA). Perhaps the researchers can find another reference for Asia, South America etc, or modify the wording. Ditto lines 51-52 that are footnoted to articles for studies in the US and Australia. If the articles themselves make this claim, maybe quote them – otherwise the reader may be left wondering.
Response 1: Thank you for raising this. We’ve added the following to expand upon this.
Lines 43-47 Companion animal relinquishment is a worldwide problem, for example every year nearly 4 million dogs are estimated to enter rescue centres in the United States (US) [2], more than 100,000 dogs enter rescue centres in Spain [3] and over 200,000 admitted to rescue centres in Australia [4].
Lines 53-57 Problematic behaviours displayed by dogs are one of the most frequent reasons for relinquishment worldwide. Carter and Taylor [13] found that dogs in the Sunshine Coast region of Australia were primarily relinquished due to behavioural reasons (15%) and in a US based study Kwan and Bain [14] found behaviour problems played at least some part in 65% of owner’s decision to relinquish their dog.
Point 2: Parag lines 62-74 – do we know why owners do not seek advice? Is it because there is none readily available, or the owners do not know where to receive this advice. I make this comment because hubby and I have always had rescue dogs and if we had had behavioural problems we would not have known where to obtain advice (Australia), apart maybe from private dog trainers.
Response 2:
Lines 77-78. Have added a couple of references which suggest that many owners are unaware of services available to help them.
Studies have suggested this could be because pet owner knowledge of the services available to help them retain their pet may be limited [18,21].
We also feel this is an interesting topic of further research and hope to do more to investigate this as part of our future work.
Point 3: Lines 87-89 “This study aims to investigate whether the presenting behavioural relinquishment reason can help to predict if an owner will accept behaviour advice to support them in retaining their dog at the point of relinquishment request.” Perhaps this can be reworded to be clearer. Also if you are saying that the reason for relinquishing a dog and its links to behavioural problems, may be an indicator whether the owners will accept advice, then it may be worthwhile to consider why owners do not seek advice – is it merely the dog’s behaviour or other reasons.
Response 3:
Have considered and reworded to hopefully be clearer, see lines 94-96.
This study aims to investigate whether the behaviour issue driving the relinquishment request can predict if an owner will accept behaviour advice to support them in retaining their dog.
Point 4: Lines 87-99 – Great! Ethics approval has been given.
Response 4: Yes, we have provided reference from Hartpury ethics board in the manuscript.
Point 5: I like the design of the project – very thorough.
Response 5: Thank you
Point 6: Lines 332-339 make a very good point.
Response 6: Thank you.
Point 7: Lines 401-403 “Some owners acquire a dog on an impulse [50] so providing advice to help well thought out decisions to be made about acquiring a dog may help to reduce some causes of relinquishment” – yes! As I was reading this article, I wondered about the human element and its contribution to the issues raised in this article. And totally agree with the conclusion around lines 428-430 about future research.
Response 7: It’s very interesting. So many different elements to consider and we hope to be able to explore these further in our future work.